

# Recession or resilience? Long-range socioeconomic consequences of the 17th century volcanic eruptions in the far north

Heli Huhtamaa[1,2], Markus Stoffel[3,4,5], and Christophe Corona[3,6]

[1]Institute of History, University of Bern, 3012 Bern, Switzerland
[2]Oeschger Centre for Climate Change Research, University of Bern, 3012 Bern, Switzerland
[3]Climate Change Impacts and Risks in the Anthropocene (C-CIA), Institute for Environmental Sciences, University of Geneva, 1205 Geneva, Switzerland
[4]Department of Earth Sciences, University of Geneva, 1205 Geneva, Switzerland
[5]Department F.-A. Forel for Environmental and Aquatic Sciences, University of Geneva, 1205 Geneva, Switzerland
[6]Geolab, Université Clermont Auvergne, CNRS, 63000, Clermont-Ferrand, France

**Correspondence:** Heli Huhtamaa (heli.huhtamaa@hist.unibe.ch)

**Abstract.** Past volcanic eruptions and their climatic impacts have been linked increasingly with co-occurring societal crises – like crop failures and famines – in recent research. Yet, as many of the volcanic cooling studies have a supra-regional or hemispheric focus, establishing pathways from climatic effects of an eruption to human repercussions has remained very challenging due to high spatial variability of socio-environmental systems. This, in turn, may render a distinction of coincidence

from causation difficult. In this study, we employ micro-regionally resolved natural and written sources to study three 17th century volcanic eruptions (i.e. 1600 Huaynaputina, 1640/1641 Koma-ga-take/Parker, and 1695 unidentified eruptions) to look into their climatic as well as socioeconomic impacts among rural agricultural society in Ostrobothnia (Finland) with high temporal and spatial precision. Tree-ring and grain tithe data indicate that all three eruptions would have caused significant summer season temperature cooling and poor grain harvest in the region. Yet, tax debt records reveal that the socioeconomic

consequences varied considerably among the eruptions as well as in time, space, and within the society. Whether the volcanic events had a strong or weak socioeconomic effect depended on various factors, such as the prevailing agro-ecosystem, resource availability, material capital, physical and immaterial networks, and institutional practices. These factors influenced societal vulnerability and resilience to cold pulses and the resulting harvest failures caused by the eruptions. This paper proposes that, besides detecting coinciding human calamities, more careful investigation at the micro-regional scale has a clear added

value as it can provide deeper understanding on why and among whom the distal volcanic eruptions resulted in different societal impacts. Such understanding, in turn, can contribute to interdisciplinary research, advice political decision-making, and enhance scientific outreach.

## 1 Introduction

Common understanding exists that volcanic aerosol forcing has been a considerable driver of pre-industrial summer temper-

ature variability at interannual-to-decadal timescales (Robock and Mao, 1995; Sigl et al., 2015; Stoffel et al., 2015; Büntgen et al., 2020), and the climate system reactions to volcanic aerosol forcing have been studied extensively over the recent decades





(see, e.g. Robock, 2000; Cole-Dai, 2010; Timmreck, 2012). By contrast, the various distal societal consequences of volcanic eruptions have hitherto received much less attention (Oppenheimer, 2015). This generalized lack of research dedicated to societal impacts of volcanic cooling is likely related to the limited availability of published societal data, especially for the more

distant past. In fact, in many states, official statistics only reach back to the 19th or 20th century, and therefore cover a time period for which the number of large eruptions is rather limited. Based on estimated global aerosol forcing, eleven out of the 20 largest eruptions of the last 2 500 years occurred between 1108 and 1815 CE (Sigl et al., 2015; Toohey and Sigl, 2017), i.e. during a time for which published societal data is generally rather scarce. For this "pre-statistical era", various historical sources can provide alternative information on the well-being of the contemporaries. Some recent interdisciplinary studies have

employed these archival sources to focus on distal societal impacts of particular eruptions of the last millennium such as the 1108/1110 unidentified (Guillet et al., 2020), the 1257 Samalas (Stothers, 2000; Campbell, 2017; Guillet et al., 2017), the 1600 Huaynaputina (Fei et al., 2016), the 1690s unidentified (D'Arrigo et al., 2020), the 1783 Laki (Demarée and Ogilvie, 2001; Witham and Oppenheimer, 2004; Grattan et al., 2007), and especially also the 1815 Tambora (Oppenheimer, 2003; Auchmann et al., 2012; Krämer, 2015; Luterbacher and Pfister, 2015; Brönnimann and Krämer, 2016) eruptions. All these studies

highlighted that volcanic eruptions can indeed have considerable, yet diverse, distal human consequences.

Although these studies have successfully detected linkages between eruptions, climatic responses, and various human repercussions, any quantification and attribution of societal impact resulting from volcanic events has remained very challenging. This is because human societies are extremely complex systems, where existing institutions, the surrounding environment, and actions of individuals, among other factors, influence the materialization of natural hazards – like volcanic eruptions – on a

societal level (Haldon, 2016; van Bavel et al., 2020; Ljungqvist et al., 2021; Degroot et al., 2021). Therefore, any distinction of how much eruption-related climatic anomalies on the one hand, and existing socio-environmental conditions and emerging human responses on the other, explain the detected societal events remains difficult . To isolate the possible volcanic effect from other natural and man-made factors, one should conduct systematic longitudinal studies, covering multiple volcanic eruptions, and compare the societal impacts associated with different eruptions over time.

Longitudinal studies require continuous and relatively homogeneous documentation on various aspects of human life, which limits the number of potential subjects of study considerably. The history of collecting and storing information systemically is commonly related to efforts aiming to gain social and economic control and to legitimize the power of authorities (Hallenberg et al., 2008). In this respect, the taxation accounts of 17th century Swedish Realm (consisting of roughly present-day Sweden, Finland, and Estonia) provide an exceptionally detailed proxy data archive on the livelihood and socioeconomic status of the

contemporaries. To utilize the limited resources of this sparsely populated northern kingdom (Fig. 1), the Crown established an extensive, centralized administration to collect taxes from the peasantry (Kujala, 2003; Hallenberg et al., 2008). The officers of the realm kept detailed records on the quantity and quality of cultivated lands, of the taxes and other tributes collected, and of possible delays and reductions of these payments.

Besides rich documentary material, the 17th century also provides good temporal scope for a longitudinal study, as it wit-

nessed three eruptions with comparable volcanic stratospheric sulfur injection (VSSI) from the 1600 Huaynaputina (Peru), the 1640/1641 Koma-ga-take/Parker (Japan/Philippines), and the 1695 unidentified tropical  (Briffa et al., 1998; Sigl et al., 2015;





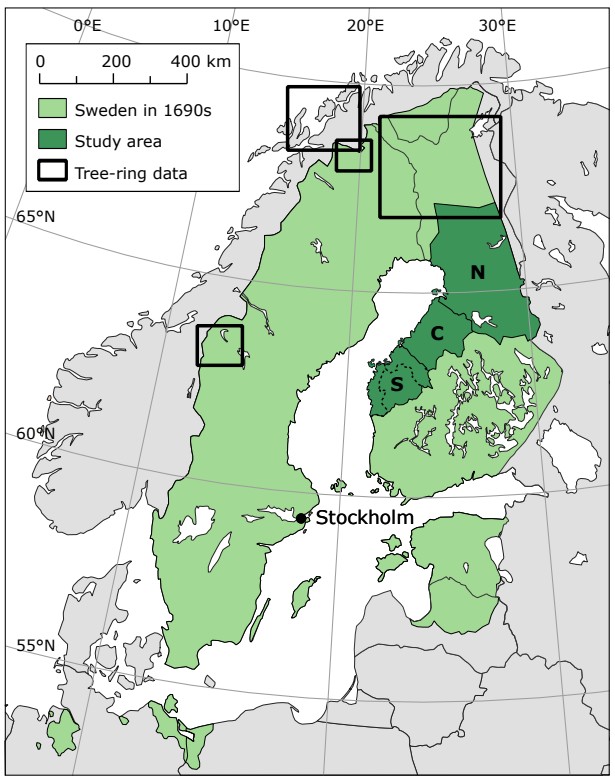

**Figure 1.** Study area, the approximate tree-ring data sample sites for the temperature reconstruction used in this study, and the sub-regions (northern, central, and three southern regions, see Table 1) referred in the text.

Toohey and Sigl, 2017; Stoffel et al., 2021; White et al., 2021) eruptions. In fact, reconstructions based on sulfate records from Greenland and Antarctica ice-cores indicate that the three eruption events are not only comparable in terms of the estimated VSSI, but also regarding global mean aerosol optical depth and global radiative forcing (Toohey and Sigl, 2017).

By integrating tree-ring and written source materials, we investigate here the micro-regional temperature response and related livelihood and socioeconomic impacts of these three 17th century volcanic events in the historical province of Ostrobothnia (present-day Finland). The objective is not limited to detecting and quantifying the societal consequences in space and time, but special emphasis is also laid on exploring the various socio-environmental factors that influenced the degree of impact. In doing so, we hope to demonstrate the potential of detailed historical research contributing to the interdisciplinary community

exploring the climatic and societal impacts of past volcanic eruptions.

## 2   Local perspective on the "Global Crisis" – Ostrobothnia in the 17th century

The 17th century CE has been described as the era of "Global Crisis", when the world was burdened by wars, rebellions, famines, and the outbreak of disease. Geoffrey Parker (2013) connected these events to deteriorating climatic conditions,





resulting from increased volcanic activity, frequent El Niño events, and decreased solar activity. Yet the climatic responses to
volcanic forcing are known to vary considerably in spatial and seasonal terms as well as across the globe (see, e.g. Fischer
et al., 2007; D'Arrigo et al., 2013; Zanchettin et al., 2013a, b; Swingedouw et al., 2017). Furthermore, broad narratives,
like the "Global Crisis", commonly fail to provide causal reasoning on the relationships between distant volcanic eruptions,
regional climate variability, and local societal consequences (Warde, 2015; Degroot et al., 2021). Previous research on the
human consequences of past volcanic events was focused largely on supra-regional crises, like famines and societal collapse.
By contrast, possible distal impacts on a local grassroots level have received much less attention. By studying the climatic
and societal impacts of the 1690s crisis in Scotland, Rosanne D'Arrigo et al. (2020) demonstrated that instead of performing
large scale analyses, a local or micro-regional spatial focus can provide deeper understanding of the spatio-temporal and
socioeconomic disparity of the human repercussions.

Consequently, the spatial scope of this study is restricted to the historical province of Ostrobothnia, located in present-
day Finland (Fig. 1). The northernmost parts (above 68.3°N) of the province are excluded from the analysis, however, as
agriculture was non-existent or marginal in the 17th century and hence the farmstead-based written documentation is limited.
The province was extremely sparsely populated in the 17th century: whereas the study area covers 111 583 km$^2$, only c. 91
150 people inhabited the area by 1695 (Muroma, 1991), of which a substantial majority were rural peasants. Their livelihood
was based on arable crop cultivation in the southern and central parts of the province, but in the north-easternmost part of
the province, agriculture consisted primarily of slash-and-burn cultivation, and in the northernmost study area people gained
their livelihood mainly from a mixture of animal husbandry and subsidiary crop cultivation (Soininen, 1974). The onset of
and the thermal conditions during the growing season were the main yield-limiting factors in this northern province, such
that solely April–August mean air temperatures can explain approximately one third of the annual pre-industrial crop yield
variations (Huhtamaa et al., 2015).

Unlike in many other parts of Europe, a majority of the 17th century Ostrobothnian peasants were freeholders. However,
as 17th century Sweden was a strictly centralized state, all land still belonged technically to the Crown, except for nobility's
properties. Thus, the freeholders had only the hereditary and usufruct rights of their holding. Yet, if a peasant failed to pay taxes
on time, the Crown took the full ownership of the farmstead, and the peasant lost its hereditary and usufruct rights (Soininen,
1974). In such a case, properties became administratively crown holdings [*kronohemman*] and their inhabitants crown peas-
ants. Additionally, new settlements, (i.e. farmsteads which cleared agricultural land from e.g. forest areas) were increasingly
categorized as crown holdings over the century. Consequently, the number of crown holdings increased over the 17th century,
and by the 1690s, the ratio between freeholder and crown peasant holdings was approximately three to one. In addition to the
freeholder and crown holdings, the study area also included varying numbers of so-called manorial peasant farmsteads (Kujala,
2003). These were peasant holdings which were located on the lands of nobility. Yet this group is excluded from this study, as
manorial peasants paid their taxes and rents to the nobility's estates and thus are largely absent in the administrative records of
the Crown.



## 3    Materials and methods

### 3.1    Temperature data

Tree-ring width (TRW), and even more so maximum latewood density (MXD) chronologies, have become the backbone of
summer temperature reconstructions and to assess the climatic impact of past volcanic eruptions (D'Arrigo et al., 2013; Stoffel
et al., 2015). To put the cooling induced by the three 17th century volcanic eruptios into perspective, we reconstructed summer
(i.e. June–August, or JJA) temperature anomalies for Fennoscandia and Ostrobothnia using multi-centennial Scots pine (*Pinus
sylvestris* L.) chronologies (MXD and TRW) from 4 sites located in northern Fennoscandia (Fig. 1) (Schweingruber et al.,
1988; Kirchhefer, 2001; Esper et al., 2012; Melvin et al., 2013; Schneider et al., 2015). Each chronology was standardized
using the Regional Curve Standardization (RCS) method as it optimally preserves multi-decadal temperature variations in the
reconstruction (Helama et al., 2017). We then transferred this record into JJA temperatures through a bootstrap linear model
using a Principal Component Analysis (PCA) calibrated against JJA land surface temperatures (1920–2000) from the E-OBS 10
min x 10 min gridded dataset (Cornes et al., 2018). For each grid point, the calibration and validation process was repeated 1000
times using a bootstrap approach. To test the robustness of the reconstruction, we employed the coefficient of determination
($R^2$ for the calibration and $r^2$ for the verification periods), RE (reduction of error) and CE (coefficient of efficiency) statistics
(Supplement, Fig. S1).

### 3.2    Grain harvest fluctuations

As the main livelihood of the majority of the study area's population came from agriculture, and specifically from crop cul-
tivation, the possible impacts of volcanic eruptions on livelihoods are studied here with grain harvest proxy data. Grain tithe
tax records are commonly used to detect annual harvest fluctuations in historical Europe (Le Roy Ladurie and Goy, 1982), and
tithes are shown to reflect relative harvest fluctuations rather well, particularly in early modern Sweden (Leijonhufvud, 2001).
In 17th century Ostrobothnia, each peasant holding was supposed to pay a share (c. 10 %) of their yearly harvest as a tithe tax
to the authorities (Huhtamaa et al., 2020). The tithes consisted of roughly equal shares of rye and barley, but in the northern
region, tithes were paid almost entirely in barley. Previously published (Huhtamaa and Helama, 2017) tithe series from South-
ern Ostrobothnia were extended to cover the whole study area. The tithe time-series were collected on a parish level, and then
transformed to indicate annual variations with respect to a pre-crisis (or pre-peak forcing) ten-year mean (the only exception
are the 1640/41 eruptions, where the time-series indicates the variation from the mean of the years 1629 and 1645–50, as tithe
data was not available for many provinces over the period 1630–40).

    The grain tithe time-series were analysed with a superposed epoch analysis (SEA) to investigate the possible volcanic signal
from the data (Robock and Mao, 1995). As the volcanic events included at least one unidentified eruption (1695), the time
series were superposed on the year of peak global volcanic aerosol forcing rather than on the year of the eruption. The dating
of peak forcing year is based on  Toohey and Sigl (2017). In addition, spatial variations and relationships between 3-year mean
harvest losses were explored at the parish level.



### 3.3 Proxy data on socioeconomic change: desertion ratio

The main economic burden for the 17th century peasant was paying taxes to the Crown. Thus, the possible socioeconomic repercussions are assessed here with tax debt records. These records are particularly appropriate to address the societal and economic consequences on a grassroots level, as majority of the people living in the studied area were peasants, their family members, or farmhands living with the peasant family. Thus the tax paying ability of a holding reflects the overall subsistence of the extended family household.

If a farmstead within the 17th century Swedish kingdom failed to pay taxes over three consecutive years, it was marked as *öde*, which means "deserted". Hence, the holding was not always deserted in a literal sense if it was marked as such (Fig. 2). Nevertheless, we will use here the literal English translation of *öde* to avoid possible anachronistic expressions, like insolvency. Besides, the category of deserted holdings included uninhabited farmsteads as well, because a holding was also marked as deserted if the peasant fled, migrated, or perished (Mäntylä, 1988). Thus, these desertion markings overall serve

as a proxy for economic recession, and in many cases, for severe societal hardships. Furthermore, desertion variations can imply changes in land tenure and social status. If a freeholder farmstead was marked as deserted, the peasant lost its hereditary right for the property, and it was released for the Crown to claim. In the best case, the peasant could continue cultivating the farm as a tenant of the Crown – i.e. as a crown peasant – without the hereditary and usufruct rights for the holding. In the worst case, the family was evicted from the farm. The situation was even more perilous when a crown peasant had tax paying

difficulties, as the authorities did not recognize their usufruct rights. Consequently, these situations commonly ended up with eviction (Kujala, 2003). In an agricultural society like that of 17th century Ostrobothnia, such evictions inevitably affected the economic possibilities and social status of a person.

The desertion material was retrieved from various accounts related to taxation, which are bound as verification, account, and land books (Fig. 2 a). These records are available in so-called Bailiffs' (up until 1634) and Provincial (from 1635 onwards)

accounts, stored at the National Archives of Finland. The number of total and deserted holdings are given in *mantal* units in the documents (Fig. 2 c). *Mantal* was a Swedish farm taxation unit, which was assigned according to the quantity and quality of the arable land of a peasant holding (Olsson and Svensson, 2010). The desertion data was collected for the seven years preceding and the seven years following peak volcanic aerosol forcing. When available, also the share of freeholder and crown peasant holdings (both, the total number of both holding types and the share of deserted holdings) was gathered. The collected

time-series indicate the average desertion ratio, that is the share of deserted holdings in relation to all holdings (based on the number of holdings existing in the year of peak volcanic aerosol forcing).

The desertion ratio data was examined with the SEA procedure similar to the grain tithe data. However, whereas the grain tithes were recorded on a parish level throughout the 17th century, the number of deserted holdings were documented on spatially wider administrative units in the years 1594–1599 and in 1602 in the sources. These wider areas are not comparable

with each other regarding the number of farmsteads (varying from 122 to 805 holdings). Thus, to avoid a possible bias towards more sparsely populated areas in the analysis, the desertion ratio SEA was performed for five comparable sub-regions (Fig 1). Additionally, the 3-year mean desertion ratios were explored at the parish level.





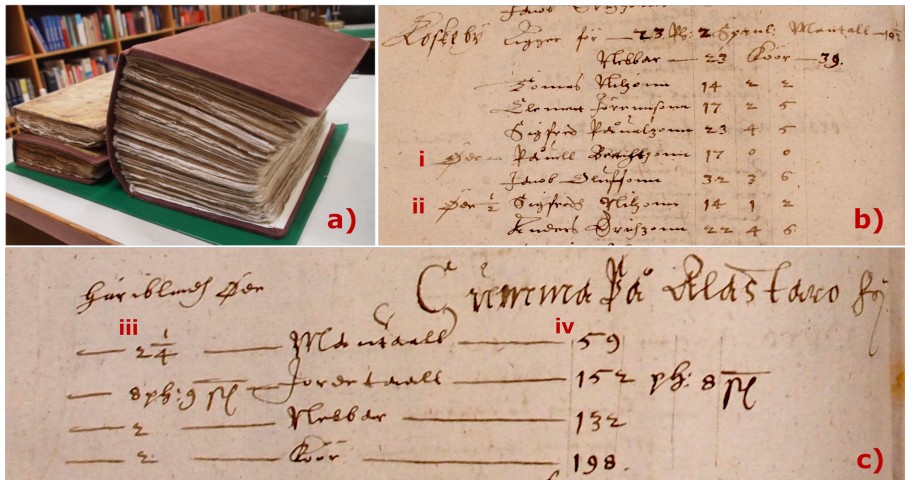

**Figure 2.** a) Early 17th century annual land, account, and verification books (from bottom left to right). b) Examples of two deserted farmsteads in the 1603 land book, which also documented the numbers of adults living in the holding (second column from the right). The first holding (i) was uninhabited, whereas the second holding (ii) had still one adult living on the farmstead. c) Example of the registers from where the desertion data for this study is gathered from: the chapel of Alastaro (Vähäkyrö) held 59 *mantal*s farmsteads (iv), of which 2.25 *mantal*s were deserted (iii).

## 4 Results

The summer temperature, grain tithe, and desertion data show clear post-volcanic signals when aligned with the years during
which volcanic stratospheric sulfur injection reached its highest values (Fig. 3, 4). Summer temperatures were reconstructed temporally over Ostrobothnia and spatially over Fennoscandia. The reconstructions are robust with a CE > 0.2 over Scandinavia and Finland; they explain more than 70 % of June–August (JJA) temperature variability approximately over the Swedish province of Lapland and Norwegian county of Nordland and 50 % over Ostrobothnia (Supplement, Fig. S1).

The mean JJA temperature reconstruction containing all grid points (10 min x 10 min) of Ostrobothnia is shown Fig. 3.
For the years following the three major 17th century eruptions, the reconstruction points to substantial cooling with -2.70°C in 1601 (with respect to the reference period 1961–1990; rank 4 in terms of cooling for the period 1500–2000), -2.27°C in 1641 (rank 11) and -1.82°C in 1695 (rank 38). The cooling in 1601 and 1641 falls within the 5th percentile (-1.98°C) of the coldest JJA temperatures reconstructed since 1500 CE. We also note, however, that very cold summers existed in the 17th century in the absence of volcanic forcing. By way of example, 1614 and 1633 were particularly cold with -2.98°C (rank 2)
and -2.89°C (rank 3), respectively. Likewise, we do not observe substantial cooling in the study region after the 1815 Tambora eruption (-0.74°C in 1816, rank 158) (Supplement, Fig. S2). The unprecedented resolution of the E-OBS dataset (10 x 10 min) also allowed comparison of the cooling reconstructed for Ostrobothnia with temperatures across the study region as well as elsewhere in Fennoscandia (Fig. 5 a, b). In order to quantify the reconstructed cooling within a context of climate variability prevailing at the time of the three volcanic eruptions, reconstructed anomalies are expressed with respect to a 31-year running





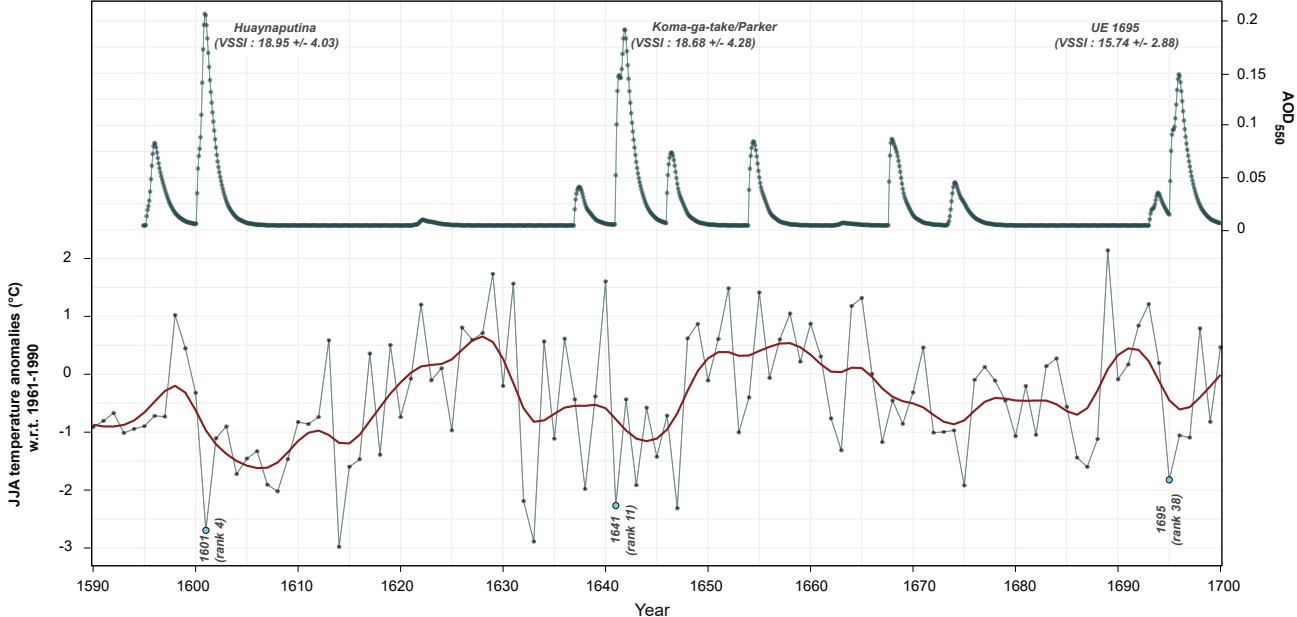

**Figure 3.** Three major eruptions are recorded in ice-core records for the 17th century in 1600 (Huaynaputina), 1640–41 (Koma-ga-take and Mount Parker) and 1695 (unidentified eruption): a) Stratospheric aerosol optical depth (SAOD) estimated for the latitude of Ostrobothnia using the Volv2k_v2 EVA AOD 500–1900 (Toohey and Sigl, 2017) dataset; b) Summer (JJA) temperature anomalies (with respect to 1961–1990) reconstructed for the period 1590–1700 over Ostrobothnia. Note that cold years followed the three major volcanic events but were also reconstructed for years without volcanic activity (e.g., 1614, 1633).

mean. For example, in the case of the year 1641 CE, a background was calculated by averaging the window 1626–1640 CE and 1642–1656 CE. The anomaly is then created by subtracting this background from the 1641 CE reconstructed temperature. In 1601, cooling is rather homogeneous across Fennoscandia - and hence also Ostrobothnia, whereas for the other eruptions, clear differences exist in the magnitude of cooling across the area. In 1641, the strongest cooling is observed north of the Arctic circle, whereas in 1695, cooling was most pronounced in southern Scandinavia (below 60°N).

When compared to the pre-crisis (or pre-forcing) 10-year mean, the tithe data suggest grain yields being less than half of the average during the year of peak volcanic forcing (Fig. 4). In 1601, grain harvest was so badly destroyed that the tithes were not collected at all. However, this year was excluded from the SEA composite as it seems unlikely that not a single grain of rye or barley was harvested – even if the contemporaries described that this was indeed the case (Ulkuniemi and Thomasson, 1975). Whereas harvest losses were the greatest in the year of peak forcing, harvest quantities remained low over the two following
years as well.

The ratio of deserted holdings increased considerably in the years after the first harvest losses (Fig. 4 b), peaking on average in the third year (+/- 1 year) following peak volcanic forcing and the first harvest failure. This is in accordance with the practice of registering holding as deserted if taxes remained unpaid for three years in a row. Yet the 15-year segments of desertion





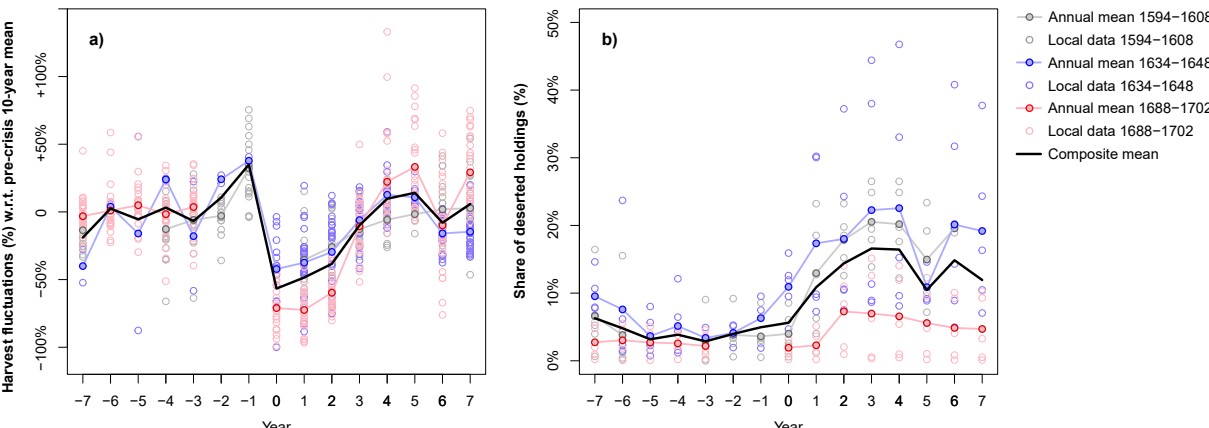

**Figure 4.** 15-year segments of composite a) grain tithe and b) desertion data indicating the grain harvest and socioeconomic responses, respectively, centred on the 17th century years of peak global volcanic aerosol forcing (1601, 1641, 1695).

ratio indicated more pronounced spatio-temporal response variations than the grain tithe series. The highest desertion ratios are

observed after the Huaynaputina and Koma-ga-take/Parker eruptions, when 20 % of the holdings, on average, were deserted from two to four years after the peak forcing. The mean desertion ratio was considerably lower (7 %) in the aftermath of the 1695 unidentified volcanic event.

As the major impact on harvest took place in the year of peak volcanic forcing and the two subsequent years, and the main increase in desertion ratios in years +2 to +4 after peak forcing (Fig. 4), we plotted mean harvest loss and desertion ratios at

the parish level covering these three-year periods, respectively, in order to gain further understanding on the spatio-temporal variability of the impacts (Fig. 5 c, d). Interestingly, no specific locality stood out. Instead, the regions hit worst in terms of harvest losses and desertion varied among the eruptions. Furthermore, the relation between harvest loss and desertion ratio varied over the different crisis periods as well (Fig. 6). Following the 1600 eruption, both harvest losses and desertion ratios were relatively high in each parish. The tithe data indicates that harvest losses were not as severe after the 1640/41 eruptions,

yet many parishes still ended up marking over 20 % of the holdings deserted. The most severe harvest losses followed the 1695 event, when mean harvest in 1695–97 was more than 56 % lower than the pre-crisis mean in each parish. However, the following desertion was moderate, as 17 out of the 26 parishes had less than 10 % of the holdings deserted (Fig. 6).

The socioeconomic impacts also varied within the peasant society. In the northern half of the study area, the accounts do not reveal whether a deserted holding was inhabited by a freeholder or a crown peasant until the late 17th century, whereas

these details are listed throughout the century in the accounts books of the three southern regions (Fig. 1). As almost all holdings were freeholder farms at the beginning of the century, all the deserted holdings were such farms as well. Due to the tax paying difficulties resulting from the hardships of the early 1600s and repeated harvest failures in 1632–1635 (Johanson, 1924), when reconstructed air temperatures were very cold in the study region – especially in 1633 and in the absence of volcanic forcing – peasants started to lose their hereditary and usufruct rights and the number of crown holdings started to rise





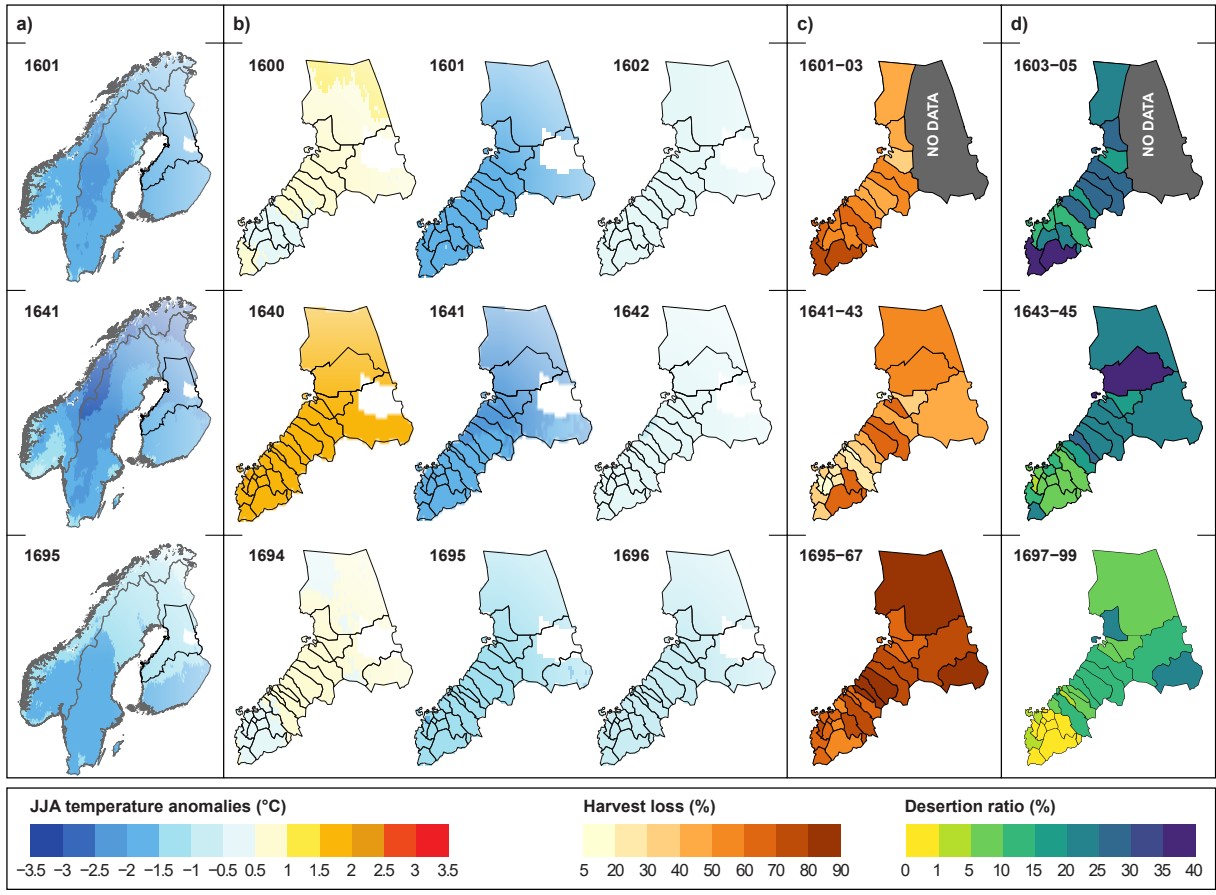

**Figure 5.** Spatial JJA temperature anomalies reconstructed over a) Fennoscandia and b) Ostrobothnia using the E-OBS 10 x 10 min gridded dataset (with respect to 31-year running mean), c) 3-year harvest loss mean (with respect to 10-year pre-crisis mean), and d) 3-year desertion ratio mean for the cooling induced by the Huaynaputina eruption in 1601 (upper row), the Koma-ga-take/Mount Parker eruptions in 1641 (middle row), and unidentified 1695 eruption (lower row).

slowly. By mid-1640s, approximately 93 % of the holdings were freeholder farms and the remaining 7 % were crown holdings. Despite crown peasants constituting less than one-tenth of the rural peasant population, their share of desertion markings in the account books is striking: from all deserted peasant farms, 48 % were crown peasant holdings. In fact, in 1645 as many as 146 out of the 171 crown holdings in the region had this marking indicating severe deprivation. The situation is similar during the 1690s crisis: by this time crown peasants constituted approximately one-quarter of the peasant population, but their

share of occupying the deserted holdings was between c. 69 % and 99 % over the years 1697 and 1699 (Table 1). Thus, the socioeconomic consequences were not equal among the rural society, but were substantially pronounced among the crown peasants.





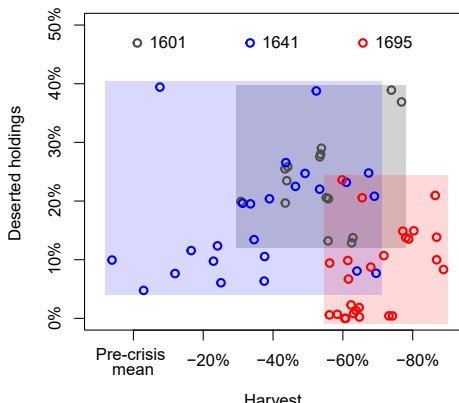

**Figure 6.** Relationship between 3-year harvest loss and desertion ratio means (data same as in Fig. 5) in Ostrobothnian parishes following the three volcanic events.

**Table 1.** Number of all holdings, the share of crown peasant holdings in relation to all holdings, the number of all deserted holdings, and the share of deserted crown peasant holdings in relation to all deserted holdings (all 3-year means) in the three southern parts of the study area (see Fig 1, dashed lines). Note that data from the central and northern parts, which constituted c. 80 % of the land area and hold c. 45 % of the farmsteads of the province, are not included in the table.

|  | 1603–1605 | 1643–1645 | 1697–1699 |
|---|---|---|---|
| All holdings (in *mantal* units) | 1963 | 1695 | 1733 |
| Of which crown peasant holdings (%) | < 1 % | 7 % | 26 % |
| All deserted holdings (in *mantal* units) | 375 | 175 | 71 |
| Of which deserted crown peasant holdings (%) | < 1 % | 48 % | 85 % |

## 5 Discussion

### 5.1 The 17th century eruptions

The 17th century saw a series of major volcanic eruptions – according to the Volcanic Explosivity Index (VEI) and using a threshold of VEI ≥ 5, between two and seven eruptions comparable to 1601, 1641 and 1695 would have occurred in the 17th century (see, e.g. Briffa et al., 1998; Esper et al., 2013; Venzke et al., 2013). By contrast, Parker (2013) even listed as many as nine VEI ≥ 5 eruptions that would have contributed to the climatic regime of the 17th century "Global Crisis" besides the 1600, 1640–41, and 1695 events. In the most recent database, Toohey and Sigl (2017) list twelve eruptions for the 17th century,

but also state that 9 of these had a limited climatic impact as their estimated Volcanic Stratospheric Sulfur Injection (VSSI) was < 5 Tg[S]. Only the three eruptions in 1600, 1640/41, and 1695 released more than 15 Tg[S], and thus likely triggered substantial cooling over the study area. In addition to the absence of a linear relation between the amount of sulfur injected to





the stratosphere and the resulting temperature cooling, one should also consider the prevailing background climatic conditions as well as eruption location and season (Robock, 2000; Zanchettin et al., 2013a), as they will critically determine the climatic

effects of a given eruption. Many past studies solely used the VEI to link climatic or societal deterioration to volcanic forcing, even if it is well established that this index does not directly indicate the climate impact potential of a volcanic eruption (Robock and Free, 1995; Gao et al., 2008). We can thus only insist that any direct attribution of societal repercussions to volcanically-forced cooling using evidence of VEI estimates should be avoided.

Whereas the cooling induced by some of the other 17th century eruptions cannot therefore be attributed easily to volcanism

alone, we evidence that the VSSI and their impact on global radiative forcing  (Sigl et al., 2015; Toohey and Sigl, 2017) was likely contributing considerably for the cold summers observed over Fennoscandia in 1601, 1641, and 1695. The strong JJA cooling shown in this study (Fig. 3), especially in 1601 and 1641, is in agreement with observations (Jones et al., 2003) and model simulations (Schneider et al., 2009), indicating pronounced post-volcanic summer cooling over most of Europe. Considering the unidentified 1695 volcanic eruption, the very cold JJA temperatures reconstructed in 1695 can be associated

to volcanic forcing if the eruption has occurred in summer or autumn 1694. Should the unidentified eruption instead only have taken place in winter 1694–1695 or thereafter, one could not exclude that the cold conditions in summer 1695 resulted simply from internal climate variability which has produced several cold extremes in the absence of volcanic aerosol forcing during the Maunder Minimum.

The grain harvest and socioeconomic responses lasted considerably longer than the volcanic-induced cold pulses (Fig. 4).

The prolonged impact on grain harvest is likely partly explained by the lack of seed grain, and partly because of the continuing unfavourable weather conditions, or the combination of both (Voipio, 1914; Muroma, 1991; Huhtamaa and Helama, 2017; Huhtamaa, 2018a). For example, the seed grain aid imports from the Baltics arrived too late for sowing in 1697 because ice clogged the Ostrobothnian harbours long into spring or as the ships were wrecked in storms (Mäntylä, 1988; Lappalainen, 2012). Also, whereas the onset of successive years of poor harvests in 1601 and 1641 coincide with the estimated peaks in

global radiative forcing and SAOD (Toohey and Sigl, 2017), the temporal correspondence between the estimated SAOD and the cold year 1695 reconstructed over Ostrobothnia is not evident (Fig. 3). Thus, we cannot rule out the possibility that factors other than volcanic aerosol forcing may have contributed to the degree of harvest losses in that particular year. Indeed, in addition to summer season temperature variability, pre-industrial crop cultivation in Finland was sensitive to winter severity and the related onset of the growing season (Huhtamaa et al., 2015; White et al., 2021). In years following long, cold and snow-rich winters,

the onset of the new growing season was badly delayed, and the crops may not have had time to ripen before the occurrence of the first autumn frosts (Huhtamaa and Ljungqvist, 2021). Noteworthy, winter temperatures in Finland are influenced partly by the intensity of westerly airflow, and the latter is linked to the modes of the North Atlantic Oscillation (NAO) (Tuomenvirta et al., 2000). The years of negative winter NAO phases are strongly correlated with cooler winter temperatures and thicker ice and snow cover, with likely impacts over on a later onset of springs and early summers due to ice–albedo feedbacks (Helama

and Holopainen, 2012). A documentary-based NAO reconstruction (Luterbacher et al., 1999, 2001) indicates that the winter (i.e. December–February) 1694–95 NAO was among the most negative (-2.56; 1st percentile) over the period 1500–2000. Furthermore, ice break-up dates from Torne river (65.84°N, 24.15°E) provide further evidence for an extremely cold winter





and spring, as the ice broke up as late as June 5, 1695 (Kajander, 1993), i.e. at the second latest ice break-up date recorded since observations started in 1693. In the year of the latest ice break-up, 1867, extreme harvest failures and hunger were

also witnessed in Ostrobothnia (Huhtamaa, 2018b). Thus, the substantial harvest losses in 1695 cannot likely be explained exclusively by the cold summer, but also by the extremely cold winter 1694–95 and the resulting delayed onset of the growing season, which may have been unrelated to the volcanic aerosol forcing.

The socioeconomic consequences varied considerable over time, space, and the peasant community (Fig. 5, Table 1). The moderate socioeconomic consequences following the 1695 unidentified eruption are rather unexpected in particular, as it is well

established that the demographic effects of this crisis were devastating. One of the most calamitous famines in European history raged in Finland in 1696–1697, when over one quarter of its population perished (Muroma, 1991). Furthermore, although the majority of the population gained their livelihood from agriculture, the severity of harvest losses did not always dictate the socioeconomic outcomes (Fig. 6). Thus, some other factors must have influenced whether the volcanic eruptions had strong or weak societal effects over different periods and locations, or among different societal groups. In order to reveal these influencing

factors, we need to investigate the socio-environmental context with high spatial and temporal precision.

## 5.2 The historical context

Following the 1601 peak in volcanic forcing, the parishes with greatest harvest losses had the highest share of deserted holdings (Fig. 5 a, d). This indicates that the crop failures had a direct influence on the socioeconomic conditions. The reason for this is likely connected to the overall stability of the society. The Swedish Realm was in great distress on the turn of the century. The

kingdom was on war with Russia, and Russian military raids caused destruction over northern Ostrobothnia (Luukko, 1945). Within the realm, King Sigismund and Duke Charles fought over the throne. These struggles materialized among the people in Ostrobothnia as well, as taxes increased and the peasants were obliged to provide maintenance and provisions for the troops passing through (Huhtamaa, 2018a). Furthermore, besides the increased fiscal burden and plundering soldiers, several crop failures emptied *the grain stores* in the 1590s. This political and military distress escalated as a peasant uprising in 1596-1597

in Ostrobothnia and as a civil war in 1598-1599 in Sweden proper (Katajala, 2002). The turmoil likely decreased fundamentally the capacity to cope with the 1601 crop failure: *institutions* for providing aid were paralyzed, *trade* was disturbed and grain could not be shipped from less affected regions (Huhtamaa, 2018a). The society was overall in a vulnerable state, and the troublesome times decreased the resilience of individuals. These factors likely explain why the harvest losses directly influenced socioeconomic conditions during the first volcanic event.

The direct spatial correspondence between harvest losses and desertion ratios weakens over the 1641 crisis (Fig. 5 c, d). This implies that some other factors may have influenced the relationship between climate variability and socioeconomic consequences. The Swedish Realm entered the Thirty Years War in 1630. The material and human resources for the war were raised mostly from the peasant society by increasing taxes and by conscripting peasants as soldiers, especially from the late-1630s onward (Villstrand, 2000; Myrdal, 2007). Thus, the main reason for the recorded tax debts in 1640s may not have been

linked to harvest failures, but to the raised overall *tax burden*. Also, likely the peasants had less grain for *storing* because the raised taxes and other financial burdens of wartime, which may have influenced negatively the individual coping capacity.





Furthermore, the wartime may have influenced also agricultural productivity, which rested on man- and horsepower (Stoffel et al., 2021). During the years 1638–1649, altogether 4 155 men were shipped from the study area to fight the war on the continent (Lappalainen, 1987). Thus, on average, almost each household was missing one adult man contributing to the *labour*, as there were approximately 4 500 peasant holdings in the area at the time. This lack of manpower may have influenced also the right timing of farm duties, which was crucial to secure a sufficient harvest (Mäkelä-Alitalo, 2003).

The harvest losses in 1695–1697 were dramatic across the study area. However, in many parishes, harvest failures had almost no socioeconomic impacts (Fig. 5 c, d). By the end of the century, Sweden had risen to one of Europe's great powers and was preparing for a war against Russia. The economy of the realm was largely financed by taxes and tributes paid by the peasantry (Lappalainen, 2012). Thus, the Crown could not afford to evict masses of peasants due to unpaid taxes, as in this sparsely populated kingdom there would not have been enough new people to take over the deserted farmsteads (Kujala, 2003). It was more reasonable for the realm to lose tax incomes for some individual years, instead of farmsteads being permanently deserted and not producing any tax payments in the future. Consequently, the Crown responded to the 1690s crisis with unusually extensive agroeconomic *relief measures* in form of *tax reductions and deferments* (Mäntylä, 1988). Furthermore, the authorities organized additional grain shipping and issued exemptions for Baltic Sea trade so as to increase the amount of imported grain (Mäntylä, 1988; Lappalainen, 2012). Yet, these actions helped people living close to seaside towns and along the main *transportation networks* (Muroma, 1991), but the measures did not reach regions with poor road networks, like the central and easternmost regions of the study area (Fig. 5 d). Besides, everyone was not entitled to the shipped grain, as it was not given out for free, but either sold or given as a loan (Lappalainen, 2012). Unlike in many western European regions, where formal *poor relief institutions* and *charity organizations* helped the poor to mitigate and cope with crop failures (Solar, 1995; van Bavel and Rijpma, 2016; Curtis and Dijkman, 2019; Dijkman, 2018; Jespersen, 2016), such formal institutions barely existed in early modern Sweden – and especially in its rural peripheries like Ostrobothnia. Thus, the rural poor were largely excluded from any institutional relief actions during the 17th century crises.

Although the socioeconomic consequences of the 1690s crisis were not substantial, they had severe demographic consequences. The extensive harvest failures triggered one of the worst famines in European history. In 1697–1698, population numbers dropped by 18 % in the southern parts and by 34 % in the central and northern parts of the study area (Muroma, 1991). The high mortality cannot, however, be explained solely by starvation per se, but by the spread of fatal hunger-related epidemics (Lappalainen, 2012). As there were no effective official poor relief institutions, food aid in the 17th century Ostrobothnia was heavily based on help that was provided by commoner to commoner. At times when a household had no food left, people left their dwellings to travel from house to house asking for food and a place to stay for the night. Thus, gaining help required mobility, which escalated the spread of infectious diseases. The differing degree of socioeconomic and demographic consequences following the 1695 event implies that there is not only one measure of "volcanic impact on society", but the intensity of the impact depends on the aspects of human life that are investigated.

When comparing the socioeconomic consequences between freeholders and crown peasants, the deprivation is substantially more profound among the latter group. As the crown holdings were commonly previously deserted or newly cleared farmsteads on the rural periphery, *geographical factors* may partly explain the differences. The location of these farmsteads, for




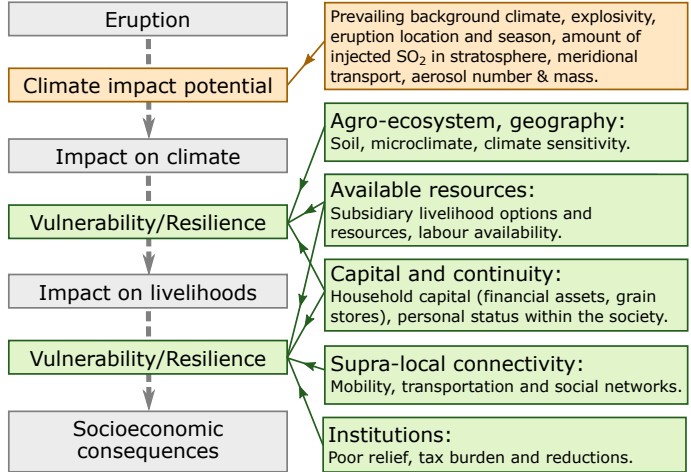

**Figure 7.** Conceptual model linking eruption events to socioeconomic consequences (left), eruption characteristics influencing the climatic impact (right, orange), and components of vulnerability and resilience (right, green), which dictate the degree of socioeconomic impacts.

example regarding *soil properties* and *microclimate*, might have simply less suitable for crop cultivation than the locations of the freeholder farms (Muroma, 1991; Solantie, 2012). Furthermore, over the century *subsidiary livelihood options*, like sea and fresh-water fishing or tar burning, became increasingly important in Ostrobothnia. The additional *financial assets* from these

345   activities may partly explain why certain areas had minor socioeconomic impacts during the 1690s crisis (Fig. 5 d). However, the rights to practice these subsidiary activities were strictly controlled by the authorities, and new licences were not commonly issued (Luukko, 1945; Virrankoski, 1973). Thus, the new crown peasant settlers may not had the same entitlement for these subsidiary sources for living as the freeholders, who have hold the rights generation after generation.

  Nonetheless, perhaps the most important factor explaining the differences in terms of socioeconomic consequences between

350   the freeholders and the crown peasants is immaterial: the continuity of *contacts*. The peasant families who had settled their holdings over multiple generations had a *social status* and respect within the community and beyond, which helped keeping trade and credit relations with burghers of the towns (Luukko, 1945; Piilahti, 2007). The mutual trust resulting from these continuous contacts were crucial for the peasants, for example, for gaining loans from the burghers to purchase shipped grain during the 1690s crisis. As an essential element of sustaining these contacts was keeping the holding within one family over

355   generations (Piilahti, 2007), newly settled crown peasant farms or holdings with earlier tax debts were greatly excluded from these arrangements. *Continuity* may thus have played a role in farm management as well, as examples from other parts of pre-industrial Europe demonstrate (Sonderegger, 2020). Peasants with hereditary rights may have been more motivated to invest into the productivity of a farm, and were likely more familiar with local challenges and ways to overcome these than newly settled crown peasants.



### 5.3  Vulnerability and resilience

The investigation of the dynamic historical context revealed how different environmental, political, institutional, and cultural factors influenced the socioeconomic consequences of the 17th century eruptions. These components (which are indicated in *italics* in the previous section) determined the societal and/or individual sensitivity to the adverse cold pulses triggered by volcanic eruptions and the capacity of the local populations to respond to and recover from these events. That is, the degree of vulnerability and resilience (Fig. 7).

The concepts of vulnerability and resilience have received critical reassessment by historians over the recent years (see, e.g. Bankoff, 2007; Haldon et al., 2020; van Bavel et al., 2020; Degroot et al., 2021). Tim Soens (2018) has noted that these two concepts are not opposing or mutually exclusive, but that a resilient societies can contain vulnerable people within. The socioeconomic consequences of the 1690s crisis demonstrates this concept: overall, the society seems to be socioeconomically resilient to the 1695 event, but closer investigation reveals one vulnerable societal group, the crown peasants, among whom the societal hardships accumulated. Likewise, a livelihood system based on climate-sensitive crop cultivation, located in one of the northernmost agricultural areas of Europe, can be considered as vulnerable in general. Nevertheless, the people who had access and entitlement to imported grain or could subside their livelihood from alternative resources were more resilient than the others. Thus, vulnerability and resilience coexists within societies, and the influence of these explaining the societal impacts of volcanic eruptions depends on the scope of our investigation.

The components which influenced the peasant society's vulnerability and resilience were neither spatially nor temporally constant. Furthermore, the effect of these components differed depending on whether, for instance, the demographic or the socioeconomic consequences are investigated. In this regard, whereas the 1601 and the 1641 events had dire socioeconomic consequences, partly due to prevailing political and military circumstances, the tax deferments during the 1690s crisis moderated the ensuing socioeconomic effects. When viewed form a socioeconomic perspective, the authorities' tax relief actions can be seen as a successful institutional coping mechanism. However, when looked from a demographic perspective, the measures that were directed only to the more advantageous segment of the peasant society had catastrophic consequences. The landless population and the less advantageous crown peasants that were excluded from the relief measures faced hunger first, and related disease epidemics started to spread, reaching eventually also the wealthier freeholders – as pathogens do not select their host by a societal status (Mäntylä, 1988). Thus, although the authorities' relief actions increased socioeconomic resilience, it also increased the vulnerability to epidemic disease.

The model (Fig. 7) was developed from the so-called impact-order model used to detect societal consequences of the 1815 Tambora eruption (Krämer, 2015; Luterbacher and Pfister, 2015). The identified components of vulnerability and resilience, however, are relevant to the case study presented in this paper. Likely, alternative components will be identified if similar models are created to study the distant socioeconomic consequences of large eruptions within different societies. Nevertheless, it is also likely that some of these components are the same regardless of time and space. For example, similar to Ostrobothnia, in Scotland, poor soil quality, lack of poor relief institutions, and socio-political distress made the society more vulnerable to




the 1690s crisis (D'Arrigo et al., 2020). Consequently, further micro-regional research focusing on different eruption periods among different societies is needed, if the model should be improved towards a more universally applicable one.

## 6  Conclusions

Historical examples can help to assess possible long-range societal impacts of future explosive volcanism and thus inform contingency planning (Oppenheimer, 2015; Riede, 2019). Additionally, past examples can advise policy makers on the unintended adverse human consequences of controversial climate change mitigation options, such as solar radiation modification (Kostick and Ludlow, 2020). Whereas archaeologists have recently developed frameworks to investigate societal consequences of past

eruptions (Riede, 2016, 2017, 2019), such theoretical work has not been carried out within the discipline of history. However, historians are working with abundant and detailed documentary sources, which enable multi-decadal, or even multi-centennial investigations with high spatial and temporal precision. Above all, historical research is able to detect various material, institutional, and cultural changes, and to estimate how these influenced the degree of vulnerability and resilience of societies to cope with volcanic cooling over time. This understanding can, in turn, contribute to interdisciplinary climate change and vol-

canological research, provide long-term perspectives to help political decision-making, and enhance scientific outreach (Riede, 2019).

Various natural and written sources, like tree-ring data, grain tithe accounts, and tax debt records in this study, enable rigorous investigation of the climatic and societal impacts of past volcanic eruptions at a given location. By applying this micro-regional approach, we evidenced that the eruption-climate-society causalities are not as straightforward as they might appear at a first

sight.

Here, we demonstrated that the summer temperature cooling following the three major distal 17th century eruptions had significant agricultural and socioeconomic impacts on contemporary Ostrobothnia. The eruptions of Huaynaputina in 1600 and Koma-ga-take/Mount Parker in 1640/41 coincide with the coldest decades of the 17th century. Whereas we reconstruct marked cooling between 1601–1610 and 1641–1646, this cooling should not the ascribed to the volcanic eruptions alone but

regarded in its climatic context with deteriorating climatic conditions of the Maunder Minimum. Thus, the societal impacts observed during these periods should not be solely attributed to volcanism either. Moreover, the relationship between harvest failures and socioeconomic repercussions was not direct – despite the fact that local populations gained their main livelihood majorly from climate-sensitive agriculture. Although the volcanic cold pulses in 1601 and 1641 – and to a lesser degree in 1695 – may have acted as triggers, the answers to the questions of why, where, and among whom the situation escalated into

a crisis are far more complex. We found that the varying degree of vulnerability and resilience, which were influenced by the prevailing agro-ecosystem, resources, capital, physical and social networks, and institutions, determined whether the eruption-related cold pulses had a strong or weak effects on human life. Importantly, these components were not mutually exclusive. For example, the same components that increased the peasant society's socioeconomic resilience to cope with harvest failures likely increased the vulnerability to hunger-related disease epidemics. Additionally, the policies on receiving grain aid helped

the wealthier freeholder peasants with good social networks, whereas the same practices made the less advantaged crown





peasants more vulnerable to crop failure. Moreover, we found that many components that obviously influenced the degree to which societal impacts were felt are difficult to quantify. For example, political stability, continuity of personal relations, and institutional practices were such factors in 17th century Ostrobothnia. Thus, understanding the distal societal consequences of past volcanic eruptions requires both quantitative and qualitative assessments.

Previous research has focused mainly on detecting famine and societal collapse following past volcanic eruptions, with limited critical assessment on the differences between coincidence and causation. Although such examples can be captivating, lessons from pre-modern large-scale crises might be difficult to adapt to today's societies. A focus on vulnerability and resilience, and how these influenced the dynamics how eruption events materialized on the grassroots level, may provide more applicable lessons. Needless to say, history cannot predict future. Yet the past can provide tangible examples, and consequently
advice decision-makers on the various long-range human consequences of future volcanic eruptions.

*Data availability.* The tree-ring datasets used in this study are freely available on the NOAA repository https://www.ncdc.noaa.gov/data-access/paleoclimatology-data/datasets/tree-ring; the E-OBS datsets can be downloaded at: https://www.ecad.eu/download/ensembles/download.php. The original archival material for the tithe and desertion data are available in the National Archives of Finland Digital Archives http://digi.narc.fi/digi/?lang=en_US.

*Author contributions.* H.H. was responsible for the initial study design, historical data collection, analysis and interpretation, and manuscript preparation, C.C. and M.S. performed the tree-ring reconstruction and climatic analyses. All authors have worked on the manuscript and approved the submitted version.

*Competing interests.* The authors declare having no competing interests

*Acknowledgements.* This research benefited from the participation in the Volcanic Impacts on Climate and Society (VICS) and the Climate
Reconstruction and Impacts from the Archives of Societies (CRIAS) working groups of the Past Global Changes (PAGES) project. The research work of C.C. and M.S. was supported by the Swiss National Science Foundation (grant no. CRSII5_183571), and the initial archival work of H.H. was supported by European Research Council sponsored project 'Coordinating for life' (grant no. 339647).





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
