# Peer review of "Recession or resilience? Long-range socioeconomic consequences of the 17th-century volcanic eruptions in northern Fennoscandia"

_Climate of the Past, 2021_

## Author Response (AR1)

Dear Francis Ludlow (the handling editor),
Dear Joseph Manning, Timothy Newfield and Katrin Kleemann,

Thank you very much for your helpful suggestions and encouraging comments on our manuscript. We will provide our final comments on the reviewer comments below. Our responses are indicated in *italics* with running numbering.

In addition to the suggested revisions, we updated the figures 4a, 5c and 6 regarding the tithe data. In the pre-print version of the manuscript, all other tithe series were indicating the change from 10-years pre-crisis mean, but the 1640/41 eruption the change from 1629 and 1645-50 mean. This was due to the fact that the Finnish archives were missing the Ostrobothnia tithe data from 1630 to 1640. Yet, as the COVID-19 travel restrictions were lifted while the manuscript was under the review, I was able to travel to Stockholm and look for the missing data at the archives there. And, indeed, the missing Finnish tithe data could be found at the National Archives of Sweden (*Länsräkenskaper*, *Norrlands län*, archival signum: SE/RA/5511/5511.27). Thus, in order to make all the three case studies as comparable regarding the data as possible, we decided to update the 1640/41 event to correspond the other case studies. Nevertheless, this update did not alter the results presented in the pre-print version of the manuscript.

Heli Huhtamaa, also behalf of all co-authors

--
RC1

This is an important paper that presents human and natural data that assesses the impact of three 17th century eruptions: Huaynaputina in 1600 (southern Peru), the double eruption of Koma-ga-take (Japan) and Mt Parker (Philippines) in 1640/41 and a hitherto unidentified eruption in 1695 (UE 1695). Volcanic eruptions can provide discrete windows onto a particular society, its vulnerabilities, and its responses to precipitation and temperature shocks in short time scales. The impact of eruptions must always be assessed against the background climate state, natural variabiilty, and the structure of the particular society. Therefore highly resoved historical data must be integrated with climate proxy data. This article does just this, and presents novel historical (tax records) and climate data (tree rings) from Ostrobothnia (Finland).

Although the region supported a very small popualtion (ca. 91-150 persons), it is the basic method here, assigning historical causation with respect to short term climate shocks in a socio-economic system with high spatial variabillty, that matters. The Abstract concisely conveys the paper's arguments and data used, the paper is well written and the arguments are very clear.

*-- #1 -- Thank you very much for these kind words. Considering the population number (ca. 91 150 persons, – not from 91 to 150 persons), we replied about this matter already on 15 December 2021.*

Since both the location and the timing of an eruption matters a great deal perhaps someting more can be said here. The use of the term "Recession" in the Title and in the paper could perhaps be changed. I am not sure that "recession" used in an economic sense is the right one here, especially in a region with a very small population. The basic point, rather, is that

certain parts of the population and certain regions were more vulerable to the shocks than others were.

*-- #2 -- We agree that the term "recession" can be somehow problematic. We have now added the word "household" proceeding the term "recession" in our definition (p. 6 in the revised MS). However, considering the population number, the number is 1000 times larger than the reviewer initially thought. In fact, the data we present in the manuscript includes every peasant farmstead in the whole province of Ostrobothnia. Thus, based on the evidence presented (for example) in Figure 4b on the percentage of farmsteads that were unable to pay their tax debts, perhaps the term household "recession" can be justified here? Furthermore, we consulted the handling editor about the matter. Consequently, we left the term "recession" in the title.*

With respect to the shocks to grain production, I wonder if something could be said about grain storage. One might expect that the ability of households to store grain for a year or two could mitigate a short-term shock.

*-- #3 -- This is very important comment. However, unfortunately, the written sources from 17th century Finland do not really capture detailed information on the peasants' household-level grain storage capacities. Also previous research do not provide much insight on the matter. Only in the case for 1690 southern Ostrobothnia we have some information on grain storage. We have now included in the manuscript on page 14.*

On land abondonment, are there other factors that can be treated?, e.g. a lack of heirs might also result in state seizure of property.

*-- #4 -- Previous research suggests the lack of heirs was not an issue in the 17th century Ostrobothnia. Instead, the situation was quite the opposite (see, e.g. Mäntylä 1988 refered in the MS). Thus, we did not add this matter as a potential factor explaining the differences.*

Figure 7 might be rethought, a more robust coupled natural-human system model with feedbacks might convey other aspects discussed in this fine study, although I take the point that here, the impact of an eruption on society is mediated by many other factors that must be considered in detail, and only examples that have highy resolved historical and climate data integrated into the analysis allow us the ability to assess how large eruptions impact societies, which, in turn, will allow policy makers to better plan for future eruptions (and potentially the impact of geoengineering).

*-- #5 -- Thank you for these insights. Indeed, we fully agree with the reviewer: more robust models might be more helpful to address similar issues regardless of the place or time. Yet, on the other hand, such detailed case studies need to be produced before one can create more general models. At the same time, none of the large eruptions of the last millennium has had the "same" effects on climate, neither in terms of amplitude nor in spatial terms, probably as a result of differing initial conditions or varying states of modes of natural climate variability. Thus, we decided to keep figure 7 as it is. Hopefully, further studies can identify an array of different socio-environmental components that could be described/assessed in more detail in the future. With such insights from different regions and time periods, we are more equipped to draw the suggested more robust model – which can be utilized also within the field of policy making.*

--

CC1

This is a very strong paper. The initial framing of it is very sharp, so too the discussion, particularly 5.2, and the figures are wonderful, especially figure 5. I agree fully with the authors that papers connecting climate cooling associated with large eruptions with socioeconomic crises are increasingly common, but almost always interregional, hemispheric or global in perspective. That is, as the authors stress, a major issue. Do those macro-scale papers ever establish causation or simply correlation? Sometimes one wonders if those papers even establish chronological *and* spatial correlation. What effects does volcanic cooling have and how do those effects differ between cultures, economies and societies? These are important questions asked here with consideration of three (!) large early modern / seventeenth-century eruptions in a single region of Finland. Often, I feel, such a narrow geographical scope is perceived to be a shortcoming — that is unfortunate, as we need more micro studies precisely like the ones this paper provides. On that note the title of the paper could better reflect the paper's local perspective; "Far North" is too wide.

The authors blend tree-ring, tax debt and tithe data to probe how three eruptions were felt by people on the ground in one region in the far north. They are assisted by the outstanding resolution of the E-OBS dataset the authors use; together these lines of evidence make for many intriguing observations about spatial heterogeneity of cooling events following each eruption studied (and makes one think twice about arguments made earlier about earlier eruptions) and about how we identify chains of causality, linking eruptions to societal crises. This care and detail is most welcomed. In short, this is a paper that makes novel and useful contributions. It is state-of-the-art in its approach and it advances the study of past climate-society linkages. It is very well suited to Climate of the Past.

*-- #6 -- Thank you so much for the positive feedback! We have now changed "far north" to "northern Fennoscandia" in the title.*

Lesser, though sometimes still significant notes:
- "Based on estimated global aerosol forcing, eleven out of the 20 largest eruptions of the last 2 500 years occurred between 1108 and 1815 CE" — this is an odd sentence. Why point specifically to this period? Considering the sentence that precedes this one, I would instead simply say that "Based on estimated global aerosol forcing, 23 of the 25 largest aerosol forcing eruptions of the last 2,500 years occurred before 1800, in the pre-statistical or pre-instrumental period."

*-- #7 -- We have now revised the sentence as suggested.*

- 'distal' is used often in the top paragraph on page 2. It could occasionally be replaced with 'far-flung'

*-- #8 -- Thank you for noting this out. We have now replaced the word 'distal' with 'far-flung' and 'far away from the eruption location' on page 2.*

- Perhaps reframe "To isolate the possible volcanic effect from other natural and man-made factors, one should conduct systematic longitudinal studies, covering multiple volcanic eruptions, and compare the societal impacts associated with different eruptions over time." *as* "To isolate possible volcanic effects from other natural and human-made factors, systematic longitudinal studies are recommended, that is, studies that span multiple volcanic eruptions and compare societal impacts associated with

different eruptions over time." While I agree with this point (reworded or not) about longitude studies, I think the period studied cannot be overly long, as societies and cultures evolve, sometimes quickly — perhaps one premodern century, but not half a millennium. The authors might note this.

*-- #9 -- Thank you for the suggestion. We have now revised the sentence as proposed. In addition, added a short comment on the appropriate temporal length when looking historical societies (p. 2, l. 44–46)*

- In the paragraph starting on line 45 on page 2, the authors would do well to note the importance of designing a longitudinal study where local high-resolution data for a climate signal that directly affected plant / crop growth are available. That (the overlap or near overlap and the proxy been very agriculturally relevant) is far too often overlooked in climate-society studies, and in this study those data are actually available — amazing.

*-- #10 – Thank you very much for this comment. We have now added a comment on the overlapping high-resolution data on p. 2.*

- Paragraph starting on line 90 of page four, a hyphen is needed between "17th" and "century Sweden/Finland", so too line 106 on page five and line 137 on page 6, etc

*-- #11 -- We have now added a following hyphen if the "17th century" phrase describes a noun throughout the manuscript.*

- Lines 177-178 of page 7, the authors might also note that cooling of the 1690s, at least in some NH regions, preceded the eruption of 1695, this seems to come up only in the discussion

*-- #12 -- We decided to focus on the section four on presenting our own results only and provide supplementary/controversial evidence from related studies in the "Discussion" chapter in the section five.*

- Is the crisis of the 1690s dated in this paper as starting in summer 1695 or earlier? That could be discussed especially on lines 210-212 on page 8 and perhaps in regards to Figure 6 (which does not of course show markedly cool winters).

*-- #13 -- The tree-ring and tithe evidence presented in the paper suggests that the onset of the volcanism-related crisis of the 1690s started in summer 1695 in **Ostrobothnia**. However, we added a notion that a dry summer of 1693 might have contributed to the crisis of the 1690s (page 12). Furthermore, we raised the matter of extremely cold (non-volcanic-induced) winter temperatures prior the summer 1695 contributing to the crises on page 12.*

- Lead sentence of section 5.1 is confusing and I would delete it. Why reference the VEI? Or introduce the complexity (frequent misuse on the part of historians) of using it? It has not been previously mentioned in the paper and as the authors know well, the VEI and Toohey/Sigl databases are not exactly comparable. I would at most just note that scholars have long thought many major climate-impacting eruptions occurred in the seventeenth century, initially using the VEI, which has been problematized, and now Toohey/Sigl. Perhaps do note erroneous previous attempts to discern climate

impacts from the VEI, but if a paragraph needs to be cut, this is the first paragraph to cut.

*-- #14 -- Thank you for raising this matter. Indeed, we agree that the VEI parameter has been misused. However, we have to disagree here about the fact that the use of VEI has been clearly problematized in previous research – especially among us historians. Thus, we felt the importance of clearly stating that societal impacts to volcanically-forced cooling using evidence of VEI estimates should be avoided. Please, see also our answer #20 to the third reviewer and the preceding comment.*

- Why are some terms italicized on page 13?

*-- #15 -- We initially decided to use italics to indicated the components that determined the societal and/or individual sensitivity (Figure 7). However, we agree that this practice can be misleading, as we use italics also with non-English vocabulary. Therefore, we highlighted the components in question with bold font instead.*

- Where was the grain imported grown? Those regions were not affected by the eruptions? I would be clearer on that (line 320 page 14)

*-- #16 -- Thank you for noting this out. We added now a comment on the export regions on page 14.*

- Missing, I felt, in 5.2 was agricultural technology. It was discussed earlier, but in brief. Were there no advancements or marked changes across this period in the region that could have made certain subregions more resilient or more vulnerable?

*-- #17 -- Overall, there was not that much changes in agricultural technology in the 17th-century Ostrobothnia and precious research has considered the period being even 'stagnated'. We added a mention of this in the revised manuscript (p. 4)*

- I am not certain the policy section of the conclusion is needed or works. Is there a way to improve it? What steps might we have to take to make a study like this policy-friendly or more easily applicable to policy makers? That said, the paper offers a lot already, it may not need these policies linkages. If it does, the micro focus, overlap of discrete evidence types, and the nuances in the historical social contextualization the paper offers are what make this paper more usable in the policy world compared to the macro studies that are so common and popular.

*-- #18 – Thank you for this insight. We have now revised the conclusions within these lines.*

- Once the authors seem to slip: from the abstract, "These factors influenced societal vulnerability and resilience to cold pulses and the resulting harvest failures caused by the eruptions." — That eruptions simply cause harvest failures is an overly simplistic statement that this paper, and Figure 6, in fact disproves (it also seems at odds with the sentence before this one in the abstract). I would change to something like "These factors influenced societal vulnerability and resilience to cold pulses and associated harvest failures."

*-- #19 -- Thank you for noticing this. We have revised the sentence as suggested.*

--

RC3

**Summary/General remarks:**

The paper of Huhtamaa et al. analyzes the long-range impact of three large, tropical volcanic eruptions in the seventeenth century (the 1600 Huaynaputina, 1640/1641 Koma-ga-take/Parker, and 1695 unidentified eruptions) by narrowing down on one specific region, in this case on Ostrobothnia in today's Finland. As these eruptions took place far away from Finland, the authors analyze the distal societal consequences these volcanic eruptions have had. The source materials for this paper are natural and written records, consisting of tree ring data and taxation / grain tithe data.

Studying the distal societal teleconnections of volcanic eruptions has been a lesser focus of scholarship so far, mainly because less data is published for this kind of analysis, particularly for the premodern era. In this sense, the paper by Huhtamaa et al. is an addition to the scholarship in this field: it showcases that tropical volcanic eruptions can have different impacts on regions far away from the volcano, which largely depends on a variety of factors, including the general climatic regime during the time, as well the socioeconomic and political conditions, all of which influence societies' vulnerability and resilience.

In the past, studies that analyze the aftermath of large volcanic eruptions and their impacts have often focused on famine and societal collapse and chose a supra-regional or hemispheric focus, which did not always address the question of causation or coincidence critically enough. By selecting a more localized regional focus, the authors of this paper address this issue.

The authors argue convincingly why they chose these three volcanic eruptions. Although recent studies (Toohey and Sigl, 2017) have identified twelve significant eruptions for the seventeenth century, only three eruptions produced significant volumes of sulfur (>5 Tg of sulfur). Huhtamaa et al. argue for relying on the estimated Volcanic Stratospheric Sulfur Injection (VSSI) rather than the Volcanic Explosivity Index (VEI), which is still often used in similar studies (lines 230-244). This is a very relevant argument that should receive more attention in future studies on volcanic eruptions.

*-- #20 -- Thank you for this comment. We agree with the sentence above. Therefore, we decided to leave the notion about the misuse of the VEI estimates (please, see also our response #14 to the second reviewer).*

Similar to the local or micro-regional spatial focus adopted by the study by D'Arrigo et al. (2020) on Scotland in the 1690s, which is also mentioned in the paper, Huhtamaa et al. employ a regional focus on Ostrobothnia that looks at three volcanic eruptions that occurred during the seventeenth century and show that this approach can provide a deeper understanding of the spatio-temporal and socioeconomic consequences. However, in contrast to the paper by D'Arrigo et al., the approach followed by Huhtamaa et al. allows for a comparative perspective. This is useful as they are able to show that the effects of these three eruptions and the socioeconomic response differs over the course of the seventeenth century, and they illustrate convincingly that various factors are responsible. To name only a few examples that are mentioned in the paper: For instance, the Maunder Minimum created

different climatic conditions during the first two eruptions (lines 413-416), the Thirty Years' Wars influenced the tax burden and military circumstances during the second eruptions (lines 302-306), and with the third eruption, tax deferments limited the desertion rate (lines 317-321).

The paper addresses scientific questions that are within the scope of CP and fit well into this special issue that addresses the volcanic impacts on climate and society. The paper presents novel ideas based on taxation records that were previously published (Huhtamaa and Helema, 2017) but extended for this paper to cover the entire study area. The paper reaches substantial conclusions: The authors show that the findings of their paper could prove useful for policymakers in the present and future concerning the "various long-range human consequences of future volcanic eruptions" (line 435).

The authors outline their scientific methods, and the results are sufficient to support the conclusions. The authors also cite relevant work in the field and indicate their original contributions. The abstract provides a concise and complete summary of the paper. The study is structured well and written clearly and well.

*-- #21 -- Thank you kindly for this encouraging and comprehensive summary.*

**Questions/Comments:**

I agree with Joseph Manning (Reviewer 1) that the "recession" in the first part of the title is not ideal. Recession is only discussed once in the paper (line 145); perhaps you could amend the first part of the title or expand the discussion on recession within the paper.

*-- #22 -- After careful consideration, we decided to keep the word "recession" in the title. However, we will consult the MS editor of the mater as well (please, see also our response #2 to Joseph Manning).*

Overall, the figures used in this paper are helpful; in particular, Figure 5 is very helpful to understand which regions are affected how much by harvest failures and to visually understand at a glance which regions see how much desertion of farmsteads in the aftermath of the different volcanic eruptions. This clarifies to the reader that harvest failures and desertions in particular regions correlate sometimes, but not always.
Figure 4b: Here, I wondered how come the desertion rate went up in the first year after an eruption if a farmstead is only marked as *öde* if the taxes are not paid three years in a row? Does this stem from the consequences of the eruption or from other conditions prior to the eruption?

*-- #23 -- Yes, indeed, the desertion rates stem from the "pre-eruption" socio-environmental conditions, which are discussed in section 5.2. Nevertheless, we added a more clear notion about this matter on page 13.*

I wondered about the deserted farms: What happened to the deserted farms that are included in the percentage; will the farmers on them either be evicted or become farmers for the crown (but not inherit the farmstead to their children)? Do other people ever come and buy a deserted farm? In other words: Do deserted farms remain in the statistic over this ~95-year time frame?

*-- #24 – We aimed to include this information in the Figure 4b. The deserted farms remain in the statistics over this period, but with much less share prior the "pre-crisis" years.*

The authors mention the uncertainty for the 1695 eruption with regard to the season, as we do not yet know which volcano produced the eruption and when exactly it took place. When looking at these eruptions and their impacts on Ostrobothnia, did you observe in your analysis whether the time of year (the season) of these eruptions plays a role for the harvest?

*-- #25 -- This is an extremely interesting question. However, unfortunately, at this point we cannot say anything about the connection between eruption seasonality and the climatic & harvest impacts over Ostrobothnia. This is because 1) we do not know the location of the volcano nor the intraseasonal dating of the 1695 event; 2) the 1641 cold pulse originated (at least) from two eruptions (see Stoffel et al. 2021, doi.org/10.5194/cp-2021-148); and 3) – to our knowledge – only the 1600 Huaynaputina eruption originated from a single known eruption (although there is also contradictory evidence, see, e.g., de Silva & Zielinski 1998, doi.org/10.1038/30948).*

**Minor points:**

I would delete the "Needless to say" in line 434.
One stylistic remark: You have both curly and straight quotation marks in your paper.

*-- #26 -- Thank you for noting these matters. We will go carefully through the manuscript to remove these inconsistencies.*

Figure 6 confused me a little, mostly because of the order of the colored rings/years above the graphic, as the order doesn't correlate with the graphic directly below. I would suggest listing the three years above one another rather than next to one another and placing them either above the graphic or on the side of the graphic. (See below for a quick draft of what I'm suggesting.)

*-- #27 -- Thank you for pointing this out. We have now modified the figure llegend.*

---

## Author Response (AR2)

13.8.2022

Dear Francis Ludlow (the handling editor),
Dear Climate of the Past editorial team,

We are very grateful to Francis Ludlow for pointing out some last cosmetic matters in our manuscript. We corrected all these matters as suggested (see our responses in the following pages). Please, just let me know if you would need also as tracked version of these changes.

In addition, we are very happy to hear that our work has been selected as a highlight paper. Thank you kindly for this.

With very best wishes,
Heli Huhtamaa, also behalf of all co-authors

[revised manuscript text omitted]